# Artificial Intelligence in Cardiovascular CT and MR Imaging

**DOI:** 10.3390/life13020507

**Published:** 2023-02-11

**Authors:** Ludovica R. M. Lanzafame, Giuseppe M. Bucolo, Giuseppe Muscogiuri, Sandro Sironi, Michele Gaeta, Giorgio Ascenti, Christian Booz, Thomas J. Vogl, Alfredo Blandino, Silvio Mazziotti, Tommaso D’Angelo

**Affiliations:** 1Diagnostic and Interventional Radiology Unit, BIOMORF Department, University Hospital Messina, 98124 Messina, Italy; 2Department of Radiology, Istituto Auxologico Italiano IRCCS, San Luca Hospital, 20149 Milan, Italy; 3Department of Medicine and Surgery, University of Milano-Bicocca, 20854 Milan, Italy; 4Department of Radiology, ASST Papa Giovanni XXIII, 24127 Bergamo, Italy; 5Division of Experimental Imaging, Department of Diagnostic and Interventional Radiology, University Hospital Frankfurt, 60590 Frankfurt am Main, Germany; 6Department of Radiology and Nuclear Medicine, Erasmus MC, 3015 Rotterdam, The Netherlands

**Keywords:** artificial intelligence, computed tomography angiography, magnetic resonance imaging, cardiovascular diagnostic technics

## Abstract

The technological development of Artificial Intelligence (AI) has grown rapidly in recent years. The applications of AI to cardiovascular imaging are various and could improve the radiologists’ workflow, speeding up acquisition and post-processing time, increasing image quality and diagnostic accuracy. Several studies have already proved AI applications in Coronary Computed Tomography Angiography and Cardiac Magnetic Resonance, including automatic evaluation of calcium score, quantification of coronary stenosis and plaque analysis, or the automatic quantification of heart volumes and myocardial tissue characterization. The aim of this review is to summarize the latest advances in the field of AI applied to cardiovascular CT and MR imaging.

## 1. Introduction

Nowadays, clinical and therapeutic interest in cardiovascular imaging is continuously increasing, with growing number of exams for dedicated radiologists who are often asked to use a long workflow prior to completing a report. Technological improvement and computational power development have allowed for huge progress in the field of artificial intelligence (AI). In the medical field, radiology represents one of the most appealing areas for AI application [1].

In particular, machine learning (ML) algorithms have the ability to learn from data, improve with experience, and make predictions [2]. Deep learning (DL) is a subtype of machine learning that does not require manual data input and generates artificial neural networks, capable of learning data and creating features [3,4]. In cardiovascular imaging, the use of AI can aid the radiologists’ workflow, reducing acquisition and post-processing time, improving image quality and exam accuracy. Moreover, stratification of the risk and prognosis evaluation can be precisely assessed due to the ability of AI to analyze enormous amounts of data [5,6].

The aim of this review is to summarize the latest AI applications in cardiovascular CT and MR imaging, pointing out the prognostic value and future prospects of a powerful but still widely unknown technology.

## 2. AI in Coronary Computed Tomography Angiography

The application of artificial intelligence in the context of cardiovascular imaging comtains several implications that mainly include the assessment of coronary artery calcium (CAC) score, stenosis and plaques evaluation, FFRct and myocardial perfusion analysis.

### 2.1. Calcium Score

The CAC score is a simple, highly reproducible tool for assessment of atherosclerotic diseases calculated on unenhanced CT scans and is considered an independent predictor of major cardiovascular events. The assessment of coronary plaque burden is actually established on a semi-automatic segmentation of calcified plaques with a density of >130 HU on axial CT images [7,8,9].

The application of AI in the evaluation of CAC would be extremely helpful, reducing the time-consuming need for the radiologist to manually select lesions and providing clinicians information about prognostic stratification of patients. 

In 2007, Isgum at al. [10] used a fully automated method for the coronary calcification detection from 76 non-contrast-enhanced, ECG-gated multi-slice CT scans including 275 calcifications in the coronary arteries. The AI algorithm was able to identify coronary calcifications in 73.8% of cases and to assign patients to the correct risk category in 93.4% of cases. 

Even if CAC score is usually calculated on ECG-gated scan, visual assessment of calcified plaque burden can be performed on every chest CT [11].

Takx et al. [12] examined a cohort of 1793 patients who underwent a non-ECG-gated and non-contrast CT for lung cancer screening to assess the impact of AI for evaluation of CACS. Even though the automated system was responsible for underestimation of calcium score values in comparison to the reference standard, results showed good reliability with a weighted k of 0.85 for Agatston risk score. 

In a more recent study, Gonzalez et al. [13] used a database of 5973 non-contrast non-ECG-gated chest CT scans, of which 4973 cases were used for training and 1000 were used for testing the ability of a 3D deep convolutional neural network (CNN) to calculate the Agatston score, reaching a Pearson correlation coefficient of r = 0.93 and an accurate risk stratification in 72.6% of cases.

These results potentially extend the application of AI CAC score stratification and benefit to a patient who undergoes a chest CT scan for different clinical indications. 

Sandsted et al. [14] evaluated the performance of an AI algorithm for the quantification of the Agatston score, the volume score and the mass score compared to semiautomated CAC score. Results showed excellent correlation (Spearman’s rank ⍴ = 0.935, 0.932, 0.934) and excellent agreement (ICC = 0.996, 0.996, 0.991, respectively). Agreement in risk category assignment was 89.5% and κ = 0.919 (*p* < 0.001).

Similar results were also obtained by Wilkemann et al. [15], who assessed an automated machine-learning system with two mechanisms, a CNN with ResNet and a dense network. The machine learning-based software achieved a great correlation (Spearman’s rho > 0.969) and excellent agreement (ICC > 0.919) with the semi-automated software used as a reference. Patients were assigned to the correct risk group in 98.4% of cases.

Martin et al. [16] presented a multi-step deep learning model and tested it on 511 patients. The first step was used to identify and segment the regions, such as the coronary artery, aorta, aortic valve, and mitral valve. The second step classified the voxels as coronary calcium. The sensitivity and the intraclass correlation coefficient they achieved were both excellent, respectively 93.2% and 0.985.

Recently, Lee at al. [17] developed a new atlas-based CAC_auto system that allows to accurately identify coronary artery regions using a deep learning model. The novel algorithm was proven using three CT angiography cohort datasets and a manually generated reference standard (CAC_hand). CAC_auto system achieved great reliability for evaluation of coronary calcium score (ICC 0.99), reaching a sensitivity of 93.3%.

### 2.2. Coronary Stenosis and Plaque Analysis

Coronary computed tomography angiography (CCTA) provides excellent visualization of coronary arteries, and its application for the assessment of coronary stenosis and plaque characterization has been widely validated [18,19,20]. AI can be very helpful in the evaluation of these features (Figure 1).

Nonetheless, post-processing and image evaluation can be very time-consuming and susceptible to inter-observer variability, which may be reduced by AI-based algorithms.

Jonas et al. [21] verified the interobserver variability among expert readers for quantifying the volume of coronary plaque and plaque components on CCTA using an artificial intelligence software as a reference (AI-QCT). The study cohort included a sub-group of 232 patients of the CLARIFY (CT EvaLuation by ARtificial Intelligence For Atherosclerosis, Stenosis and Vascular MorphologY) study. Readers yielded fair consistency in quantifying total plaque volumes (Spearman’s coefficients ranging between 0.68 and 0.74), but there was a certain grade of discordance for the assessment of plaque composition (Spearman’s coefficients between 0.38 and 0.60).

Based on stenosis and plaque burden evaluated on CCTA, patients are assigned to specific category of risk according to the Coronary Artery Disease-Reporting and Data System (CAD-RADS) classification allowing to identify patients that may require further functional or invasive investigations [22].

Muscogiuri et al. [5] evaluated a cohort of 288 patients who underwent CCTA to check the performance of a deep learning algorithm based on CNN for the classification of CAD-RADS. The average time of analysis of CNN was significantly shorter compared to that of humans. Sensitivity, specificity and accuracy ranged between 47 and 82%, 58 and 91%, and 46 and 86%, respectively.

Paul et al. [23] valued a deep-learning model (DLM) trained with 10,800 curved multiplanar reformatted (cMPR) CCTA images for classifying coronary arteries on CCTA using the CAD-RADS. The results showed that sensitivity and specificity were 93% and 97%, respectively, and the negative predictive value was 97%.

Choi et al. [24] enrolled 232 patients to compare AI performance in assessing maximal diameter stenosis, plaque volume and composition, presence of high-risk plaque and CAD-RADS category in comparison with three expert radiologists. AI’s accuracy, sensitivity, specificity, positive predictive value and negative predictive value were excellent for both stenoses >70% and >50%: 99.7%, 90.9%, 99.8%, 93.3%, 99.9%, 94.8%, 80.0%, 97.0, 80.0%, and 97.0%, respectively. Patients were assigned to the correct CAD-RADS category in 228/232 (98.3%) exams.

Several studies have compared the diagnostic accuracy of artificial intelligence in the assessment of stenosis to more invasive diagnostic approaches.

Hell et al. [25] applied an AI-based software (AUTOPLAQ) to define the maximum percent difference of contrast densities (CDD) within an individual lesion to calculate the hemodynamic relevance of coronary stenosis. CDD was considerably increased in hemodynamically significant stenosis and allowed to determine the hemodynamical relevance of the lesions with a specificity of 75% and negative predictive value of 73% when compared to invasive FFR.

In a multi-center study, Griffin et al. [26] compared the accuracy of AI for the detection of >50% stenosis and >70% stenosis to quantitative invasive angiography (QCA). Results showed an accuracy ranging between 84% and 86% and high correlation between stenosis detected on AI-QCT evaluation vs. QCA (ICC = 0.73). The analysis required approximately 10 min, with a net gain in time against the hours normally necessary for readers to process images. However, AI was responsible for the detection of 62/848 (7.3%) false-positive coronary stenosis ≥ 70%.

Over the last few years, the development of dual-energy technology has significantly increased its use in cardiovascular diagnostics due to its numerous advantages [27,28,29,30].

Yi et al. [31] investigated the value of an AI-based application for detecting coronary stenosis from virtual monoenergetic spectral reconstructions (VMI) on Dual-Energy CCTA using invasive coronary angiography (ICA) as reference standard. VMI images demonstrated a non-inferior diagnostic performance in vessel analyses with conventional images, reaching a similar sensitivity (72 vs. 74%) and even higher specificity (91 vs. 86%) and diagnostic accuracy (82 vs. 80%). For plaque assessment, the accuracy of VMI ranged between 80 and 93% vs. 81 and 94% of conventional imaging.

### 2.3. FFRct and Myocardial Perfusion

Fractional flow reserve computed tomography (FFRct) is a novel and non-invasive instrument for the detection of myocardial ischemia based on computational fluid dynamics modeling techniques [32].

The development of machine learning algorithms for the estimation of FFRct could be incredibly valuable for the assessment of potential ischemic lesions.

Coenen et al. [33] used the data obtained from MACHINE consortium to validate the accuracy of ML-FFRct compared with invasive FFR as a reference standard and with FFRct derived from the classic fluid dynamic (CFD) FFRct. The authors determined that on a per-vessel analysis, the diagnostic accuracy of ML-FFRct was 78% compared to 58% of CCTA, while the patient accuracy reached 85% compared to 71% of CCTA. ML-FFRct and CFD-FFRct showed the same area under the curve (AUC: 0.84) compared to the low AUC of CCTA (AUC: 0.69).

A study by Tesche et al. [34] compared CCTA to ML-FFRct in 85 patients using QCA as reference standard, revealing a per-lesion and a per-patient sensitivity of 79% and 90% and a specificity of 94% and 95%, respectively.

In their retrospective study, Morais et al. [35] proposed the FFRct obtained by an AI-based software to improve the detection coronary ischemia. The authors analyzed images from 93 patients who underwent CCTA on two different scanners and detected good agreement between FFRct and invasive FFR which have been used as a reference standard (r = 0.73). The use of different scanners did not affect the correlation between FFRct and invasive FFR. FFRct performance was considerably higher compared to the visual classification of coronary stenosis (AUC 0.93 vs.0.61).

Tang et al. [36] assessed the value of ML-FFRct in identifying hemodynamically in-stent restenosis using invasive FFR as reference in 33 patients who underwent stent implantations. Results showed that FFRct had a great correlation with invasive FFR (ICC = 0.84).

Recently, Quiao et al. [37] performed a prospective study whose aim was to prospectively compare the outcomes of ML-FFRct with conventional anatomical CCTA or the evaluation of patients with moderate coronary stenosis. Patients were divided in two groups, depending on whether they were assigned to CCTA or FFRct group. At 90 days, the rate of ICA without obstructive disease was higher in the CCTA group (33.3%, 39/117) than that in the FFRct group (19.8%, 19/96). FFRct was also associated with a lower rate of referral for ICA (20.3% vs. 27.5%) and 1-year MACE when compared to the anatomical CCTA alone strategy (HR: 1.73; 95% CI: 1.01, 2.95; *p* = 0.04).

Myocardial CT perfusion (CTP) also demonstrated to improve the accuracy of CCTA in recognizing hemodynamically significant coronary stenoses [38,39]. 

Nevertheless, AI applications in CTP are actually limited, and only few studies have investigated its potential.

Xiong et al. [40] evaluated the capability of three ML models (Naive Bayes, Random Forest, and AdaBoost) to differentiate significant coronary artery stenosis from myocardial perfusion assessed by CCTA imaging at rest using QCA as reference. AdaBoost showed a sensitivity of 0.79 and specificity of 0.64 when compared to QCA.

Han et al. [41] developed an ML algorithm that was able to recognize myocardial perfusion deficits at rest using datasets from 252 CCTAs. The ML model showed, on a per-patient analysis, accuracy, sensitivity, specificity, positive predictive, and negative predictive of 68.3%, 52.7%, 84.6%, 78.2%, and 63.0%, respectively, and improved stenosis detection (AUC: 0.75 vs. 0.68 of CCTA evaluation alone).

Muscogiuri et al. also proposed a DL algorithm for the identification of ischemic myocardium on rest CCTA with a sensitivity, specificity, NPV, PPV, accuracy, and AUC respectively of 100%, 72%, 100%, 74%, 84%, and 96% [42]. Moreover, the application of the ML model also allowed to shorten the time of analysis compared to human evaluation (39.2 vs. 379.6 s).

## 3. AI in Cardiac Magnetic Resonance

Cardiac Magnetic Resonance (CMR) may require a long time, both for the acquisition of images and their segmentation, such as for cardiac volumes assessment. In clinical practice, the analysis of MRI datasets based on manual analysis can lead to long post-processing times and inter- and intra-operator variability.

In this scenario, AI may represent the keystone to help radiologists in their clinical workflow as well as to increase the accuracy, reproducibility, and precision of their examinations [43].

### 3.1. Acquisition Phase

CMR imaging requires accurate and precise execution of the acquisition phase by the technician in order to obtain the standard views recommended by the SCMR guidelines [44]. Since the availability of dedicated technicians is not always possible, multiple studies have been exploring the opportunity to automatise, speed up, and standardize CMR acquisition. 

In a study conducted by Lu et al. [45], a new approach based on learning-based algorithms was proposed to automate and accelerate the acquisition process. First, they manually segmented a left ventricle (LV) 3D model that was co-registered and scaled to the patient heart through a tree-based classifier, with probabilistic boosting trees. Then, the 3D model was inflated or deflated to match the specific patient heart anatomy. Other studies proposed different algorithms to obtain anatomical landmarks used for slice alignment [46,47,48,49].

Fokati et al. [50] applied a joint Multi-Scale Variational Neural Network to accelerate the reconstruction time of a prototype of balanced-Steady-State Free Precession sequence for 3D whole-heart imaging: the Free-breathing Magnetization Transfer Contrast Bright blOOd phase SensiTive (MTC-BOOST). They concluded that the proposed five-fold accelerated jMS-VNN MTC-BOOST framework provides efficient 3D whole-heart bright-blood imaging in fast acquisition (3.0 ± 1.0 min vs. 9.0 ± 1.1 min) and reconstruction time (10 ± 0.5 min vs. 20 ± 2 s) compared to conventional Compressed Sensing with concomitant reduction in flow and off-resonance artifacts. 

Even Weine et al. [51] attempted to improve the acquisition of cardiac diffusion tensor imaging (cDTI), which can provide information about myocardial microstructure. They hypothesized that the robustness of diffusion tensor estimation can be improved by incorporating spatial information and physiologically plausible priors into the fitting algorithm. Thus, they trained a CNN with synthetic data generated by a parameterized pipeline (including spatial correlations of diffusion tensors and motion of the heart). The CNN demonstrated an increase in the accuracy and precision of cDTI and a potential reduction in acquisition time. 

### 3.2. Image Segmentation

Cine-MRI provides essential information about ventricular wall motion abnormalities, myocardial thickening, and ventricular volumes [52,53,54]. Manual segmentation of cardiac volumes by tracing endocardial and epicardial contours is essential for biventricular function evaluation [55]. However, contouring is a time-consuming process, which can be even more difficult when there are individual differences in heart shape or datasets with low contrast-to-noise ratio. Additionally, inter- and intra-operator variability can still be a problem in follow-up studies [56,57]. 

Different AI models have been developed to automate the segmentation process, making it easier, faster, and more accurate (Figure 2).

During the 20th International Conference on Medical Image Computing and Computer Assisted Intervention (MICCAI), the “Automatic Cardiac Diagnosis Challenge” (ACDC) was performed to establish the best AI model for automatic cardiac segmentation. Bernard et al. tested multiple DL algorithms for the segmentation and classification task [58]. They demonstrated a 0.97 correlation score for the best algorithm, which proved to work very well for LV segmentation, but it was suboptimal for right ventricle and myocardial evaluation. 

Bay et al. tested a fully convolutional network (FCN) automated analysis trained on a large-scale UK Biobank dataset consisting of 4875 subjects with 93,500 pixel-wise annotated images [59]. They demonstrated a human-level performance on the UK Biobank dataset, although this dataset was relatively homogeneous and based on a single scanner model with a standard imaging protocol. 

A multicenter and multivendor study conducted by Tao et al. [60] demonstrated high accuracy of LV function quantification based on three CNNs with the U-NET architecture with a correlation of 0.98 and an average perpendicular distance of 1.1 mm ± 0.3 compared with manual analysis. 

Li et al. [61] proposed a DL-based cardiac MRI segmentation scheme based on three steps. First, they pre-processed the input MRI data using a standard deviation filter to detect regions of the cardiac cycle image sequence where pixel intensity varied strongly with time, and then they used Canny edge detection and Hough transform techniques to detect regions containing the heart area of interest. Second, the images were input into the ESA-Unet model network, and through the encoder self-attention module and decoder processes, a preliminary segmentation result was obtained. Third, the conditional random field was used to reprocess the segmented images and optimize segmentation boundary. This method demonstrated a good segmentation effect, which facilitates the diagnosis of clinical cardiovascular diseases, improves the efficiency, and accuracy of diagnosis.

For left atrial segmentation, Wong et al. proposed a U-Net with Gaussian blur and channel weight neural network to automatically segment the left atrial region in the MRI of a patient with left atrial enlargement [62]. Their CNN-based technique results in the segmentation of the left atrium being closer to the manual segmentation by an experienced radiologist, with a dice similarity coefficient (DSC) of 93.57%.

### 3.3. Tissue Characterization

Non-invasive tissue characterization is one of the distinctive features of CMR [63,64,65,66,67,68,69,70,71]. 

Late gadolinium enhancement (LGE) imaging together with T1 and T2 mapping techniques provide both visualization and quantification of focal or diffuse myocardial disease [72,73,74,75,76,77]. In particular, native T1 mapping techniques are able to detect increased extracellular compartment, as it happens in amyloidosis, acute inflammation, or myocardial fibrosis, as well as the presence of iron infiltration or Fabry’s disease [67,78,79,80,81]. Moreover, increased T2 mapping values are extremely accurate in assessing myocardial edema [63,82]. 

AI could be of great support for the correct evaluation of the aforementioned parameters, which are of great importance both for diagnostic and prognostic models [83,84,85].

Chang et al. [86] attempted a DL method for the automated measurement of native T1 and extracellular volume (ECV) fractions in CMR imaging with a temporally separated dataset. They demonstrated that DL algorithm successfully segmented the myocardium in 99.3% of slices in the native T1 map, and 89.8% of slices in the post-T1 map with DSC of 0.86 ± 0.05 and 0.74 ± 0.17, respectively. Native T1 and ECV showed a strong correlation and agreement between DL and the reference, and the agreement between DL and radiologists was excellent. 

AI algorithms can also be used to reduce artefacts and to improve the accuracy of native myocardial value estimation. Li et al. [87] evaluated a motion correction method for myocardial T1 mapping using self-supervised deep learning-based registration with contrast separation (SDRAP). Results showed that the AI algorithm achieved the highest DSC of 85.0 ± 3.9% and the lowest mean boundary error (MBE) of 0.92  ±  0.25 mm among the methods compared.

AI-based automatic contouring was also developed to obtain precise LGE quantification (Figure 3).

Moccia et al. [88] tested a segmentation model based on FCN for LGE segmentation and provided a DSC of 71.3%, with a sensitivity, specificity, and accuracy of 88.1%, 97.9%, and 96.8%, respectively. 

Zabihollahy et al. [89] used a cascaded multi-planar U-Net (CMPU-Net) to efficiently segment the boundary of the LV myocardium and scar from 3D LGE-MR images. Their algorithm reported a DSC of 85.14% ± 3.36% by comparing it to manual segmentations. 

On the contrary, Zhang et al. [90] proposed a method based on non-contrast cine images. They used a deep learning framework to identify the presence and extension of myocardial infarction on cine MRI by first localizing the LV, and then analysing both local and global motion features. The results obtained were promising with sensitivity, specificity, and AUC of 89.8%, 99.1%, and 0.94%, respectively, on non-contrast cine images. 

In another recent study, the same research group aimed to assess virtual non-enhancement (VNE) technology, which showed a strong correlation with LGE in quantifying scar size (r, 0.89) and transmurality (r, 0.84) in 66 patients. In addition, VNE demonstrated values of accuracy, specificity, and sensitivity of 84%, 100%, and 77%, respectively [91,92].

Sendra-Balcells et al. trained a model that can maintain a good level of performance when used to segment out-of-sample images from new hospitals [93]. The results showed that the combination of data augmentation and transfer learning can lead to single-center models that generalize well to different clinical centers which were not included during the training phase.

Atrial fibrosis segmentation also represents a key prognostic and risk factor for the onset of atrial fibrillation [94,95]. Li et al. proposed a fully automated method using a multi-scale CNN for the assessment of atrial scars [96]. They demonstrated a mean accuracy of 0.856 ± 0.033 and a mean DSC of 0.702 ± 0.071 for left atrium scar quantification. Even Cho et al. used a 3D U-net architecture using a limited dataset of LGE CMR, demonstrating a DSC value of 0.90 [97].

### 3.4. Prognosis

One of the main strengths of AI application is the possibility to better stratify the risk and identify the prognosis in patients with cardiomyopathies or those subjected to invasive treatments [83,98,99]. 

A combined approach based on the analysis of radiomic features with AI algorithms was proposed by Arian et al. [100]. They used a smoothly clipped absolute deviation (SCAD)–penalized support vector machine (SVM) and the recursive partitioning (RP) algorithm to predict myocardial function improvement in patients who had undergone coronary artery bypass grafting (CABG). The authors concluded that multiple radiomic features alone or combined in the multivariate model using machine learning algorithms provide prognostic information regarding myocardial function improvement in patients after CABG. 

Dawes et al. [101] used a machine learning survival model based on three-dimensional cardiac motion to predict the outcome in patients with pulmonary hypertension independently of conventional risk factors. 

Chen et al. [102] proposed a ML method using naive Bayes classifier for risk prediction in patients with severe dilated cardiomyopathy at one-year follow-up. Their results showed a high prognostic accuracy of the AI algorithm (AUC of 0.88) compared to 0.59 for the Meta-Analysis Global Group in Chronic Heart Failure score and 0.50 for LVEF [103]. 

In patients with repaired tetralogy of Fallot, Diller et al. trained a machine learning algorithm on external imaging datasets, determining a strong role of the right atrial median area and the right ventricular long axis strain as outcome predictors [104].

## 4. Conclusions

Currently, the vast majority of AI applications are limited to research purposes. However, in the context of constant technological development and considering the increasing demand for cardiovascular diagnostic studies, AI will necessarily acquire a fundamental role in clinical practice. In cardiovascular imaging, AI algorithms are going to improve workflows and aid radiologists in the detection of abnormalities, thus simplifying quantification and improving the final diagnostic and prognostic accuracy. Finally, some AI applications may also provide information that has been achieved only by means of constrastographic studies, which translates to even greater advantages both for economics and patients’ health.

In the near future, the application of AI may result in faster image acquisition, radiation dose reduction without loss of image quality for CT studies, and improvement in cardiovascular disease diagnosis with facilitated reporting process. Additionally, AI could potentially integrate imaging data with clinical and laboratory data in order to achieve even higher accuracy rate in patient diagnosis and management.

It is unlikely that AI will completely substitute the human contribution, since the physician supervision will always be necessary for result validation.

Currently, AI application in clinical routine has several limitations. First, every model requires to be validated by larger studies, implying the necessity to achieve a standardization of imaging protocols across different centers, to prove the real effectiveness and accuracy of AI. The incredible amount of data obtained is also leading to major challenges in terms of patients’ privacy and data protection. Second, the application of AI models in clinical practice needs to be approved by the Food and Drug Administration or the European Medicines Agency.

In conclusion, AI may represent a breakthrough in the technical innovation of imaging, not strictly limited to cardiovascular investigations, potentially providing new data in every imaging modality, ranging from X-rays to PET-CT and improving physician diagnostic confidence, by reducing the number of diagnostic errors [105,106,107].

## Figures and Tables

**Figure 1 life-13-00507-f001:**
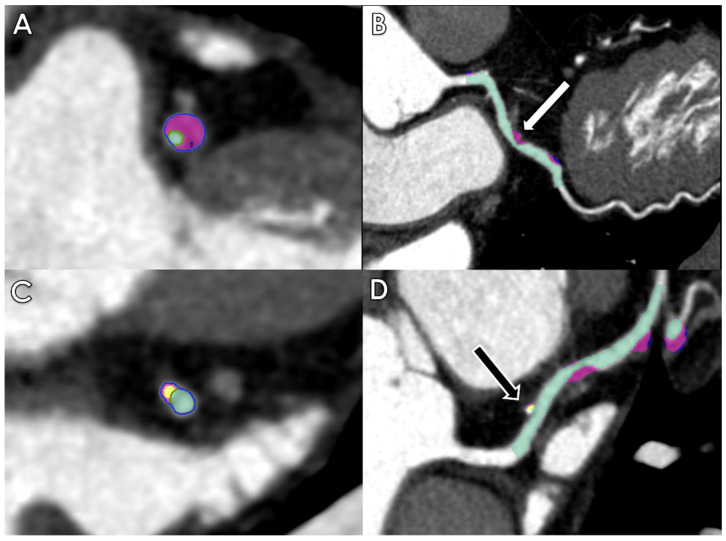
Axial and multiplanar reconstruction of quantitative plaque AI-based measurements of a fibrofatty plaque with positive remodelling determining severe stenosis of the left descending artery (LAD) ((**A**,**B**)—white arrow) and of a calcified plaque determining mild stenosis of the LAD ((**C**,**D**)—black arrow).

**Figure 2 life-13-00507-f002:**
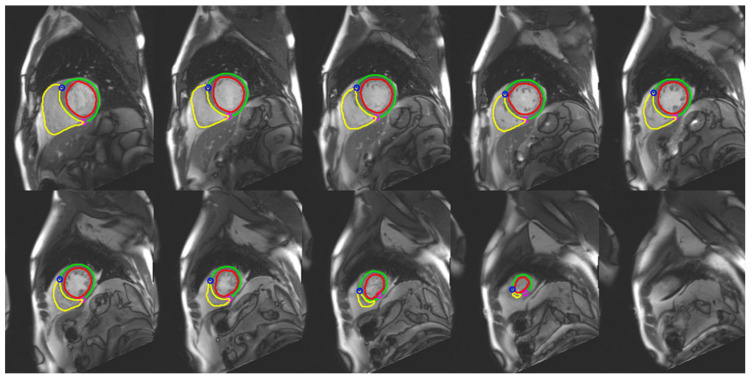
AI-based segmentation of right and left ventricles in end diastolic phase in a 55-year-old patient.

**Figure 3 life-13-00507-f003:**
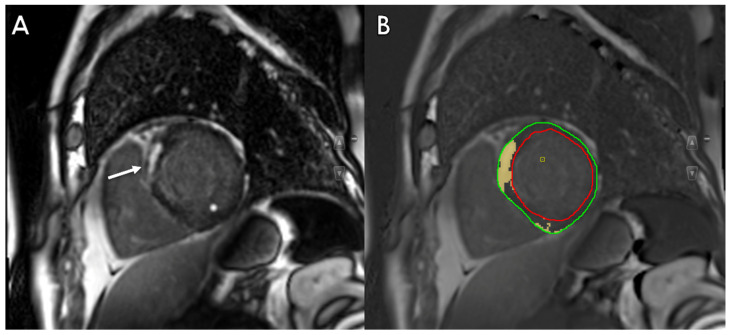
Assessment of myocardial Late Gadolinium Enhancement (LGE) in a 55-year-old patient showing areas of contrast enhancement with subendocardial and transmural distribution (white arrow—(**A**)) automatically detected using artificial intelligence (yellow-colored myocardium—(**B**)).

## Data Availability

Not applicable.

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
