# Peer review of "Artificial Intelligence in Cardiovascular CT and MR Imaging"

_life, 2023, doi:10.3390/life13020507_

Round 1
Reviewer 1 Report
My overall assessment is very high for this article! This is a great study! Authors made an excellent study! I have only a minor suggestion to make this article even better! Even PET/CT has pitfalls and may artificial intelligent will help! This article provide all the necessary information and can be cited! Tsetsos N, Poutoglidis A, Arsos G, Tsentemeidou A, Kilmpasanis A, Katsampoukas D, Fyrmpas G. 18F-FDG-PET/CT interpretation pitfalls in patients with head and neck cancer. Am J Otolaryngol. 2022 Jan-Feb;43(1):103209. doi: 10.1016/j.amjoto.2021.103209. Epub 2021 Sep 10.
Author Response
Response: Thank you, we adjusted this accordingly and added the suggested reference.
Reviewer 2 Report
The manuscript is a review article titled Artificial Intelligence in Cardiovascular Imaging.
AI has the potential to advance cardiovascular imaging by facilitating each step of the imaging process, including image acquisition, quantification, analysis, and reporting. AI can reduce human error in cardiovascular imaging applications and saved time in the clinical workflow.
This article covers two areas of cardiovascular imaging which are Coronary Computed Tomography Angiography (calcium score, coronary stenosis and plaque analysis, and FFRct and myocardial perfusion) and Cardiac Magnetic Resonance (acquisition phase, image segmentation, tissue characterization, and prognosis).
A review article is written to summarize the current state of understanding on a topic, which is cardiovascular imaging in this manuscript. Apart from the two areas that the authors included, there are more areas to be covered such as Interventional Cardiology and Echocardiography, to name a few.
How about other modalities such as nuclear cardiology (such as myocardial perfusion imaging / cardiac SPECT, or PET-CT), angiography imaging, identification of abnormalities on chest X-rays, dysrhythmias on electrocardiograms, or heart sounds using electronic stethoscopes?
What are the potential drawbacks and challenges of applying AI in cardiovascular imaging?
Future directions of AI in in Cardiovascular Imaging?
Author Response
Comment#1:
A review article is written to summarize the current state of understanding on a topic, which is cardiovascular imaging in this manuscript. Apart from the two areas that the authors included, there are more areas to be covered such as Interventional Cardiology and Echocardiography, to name a few.How about other modalities such as nuclear cardiology (such as myocardial perfusion imaging / cardiac SPECT, or PET-CT), angiography imaging, identification of abnormalities on chest X-rays, dysrhythmias on electrocardiograms, or heart sounds using electronic stethoscopes?
Response: Thank you for the important remark. Our aim was to provide an overview on the current technical advances of artificial intelligence applied to CT and MR cardiovascular imaging. As you correctly highlighted, we did not cover AI applications in other imaging techniques. We changed the title of the manuscript from “Artificial Intelligence in Cardiovascular Imaging” to “Artificial Intelligence in Cardiovascular CT and MR Imaging”.
Comment#2:
What are the potential drawbacks and challenges of applying AI in cardiovascular imaging?
Response: Thank you, for the comment. We included a brief paragraph about AI challenges in the conclusions.
Comment#3:
Future directions of AI in in Cardiovascular Imaging?
Response: Thank you, we emphasized this concept in the conclusions as kindly suggested.
Round 2
Reviewer 2 Report
The authors have satisfactorily addressed the comments.